# Robustness Assessment of Oncology Dose-Finding Trials Using the Modified Fragility Index

**DOI:** 10.3390/cancers16203504

**Published:** 2024-10-17

**Authors:** Amy X. Shi, Heng Zhou, Lei Nie, Lifeng Lin, Hongjian Li, Haitao Chu

**Affiliations:** 1Cardiovascular, Renal and Metabolism (CVRM), Biopharmaceuticals R&D, AstraZeneca, Durham, NC 27703, USA; 2Biostatistics and Research Decision Sciences, Merck & Co. Inc., Rahway, NJ 07065, USA; heng.zhou@merck.com; 3Division of Biometrics IV, OB/OTS/CDER/FDA, Silver Spring, MD 20993, USA; lei.nie@fda.hhs.gov; 4Department of Epidemiology and Biostatistics, Mel and Enid Zuckerman College of Public Health, The University of Arizona, Tucson, AZ 85721, USA; lifenglin@arizona.edu; 5Cardiovascular, Renal and Metabolism (CVRM), Biopharmaceuticals R&D, AstraZeneca, Gaithersburg, MD 20878, USA; hongjian.li@astrazeneca.com; 6Statistical Research and Data Science Center, Pfizer Inc., New York, NY 10001, USA; 7Division of Biostatistics and Health Data Science, The University of Minnesota Twin Cities, Minneapolis, MN 55455, USA

**Keywords:** oncology trial, fragility Index, maximum tolerated dose, trial design, dose finding, sensitivity analysis, early stopping

## Abstract

**Simple Summary:**

In this article, the authors introduce a new metric called the modified Fragility Index (mFI) to assess the accuracy of determining the maximum tolerated dose (MTD) in early oncology clinical trials. The mFI measures how sensitive the MTD decision is to the inclusion of a few more participants in the trial. The authors analyzed three published cancer trials and found that two trials were robust to adding more participants, indicating that the MTD estimate remained stable. However, in the other trial, the MTD estimate was more fragile and could have changed with just one or two more participants. The mFI metric helps researchers make more reliable decisions about the appropriate MTD. By considering the potential impact of additional participants, researchers can improve accuracy and confidence in dose determination, leading to better treatment outcomes for patients.

**Abstract:**

Objectives: The sample sizes of phase I trials are typically small; some designs may lead to inaccurate estimation of the maximum tolerated dose (MTD). The objective of this study was to propose a metric assessing whether the MTD decision is sensitive to enrolling a few additional subjects in a phase I dose-finding trial. Methods: Numerous model-based and model-assisted designs have been proposed to improve the efficiency and accuracy of finding the MTD. The Fragility Index (FI) is a widely used metric quantifying the statistical robustness of randomized controlled trials by estimating the number of events needed to change a statistically significant result to non-significant (or vice versa). We propose a modified Fragility Index (mFI), defined as the minimum number of additional participants required to potentially change the estimated MTD, to supplement existing designs identifying fragile phase I trial results. Findings: Three oncology trials were used to illustrate how to evaluate the fragility of phase I trials using mFI. The results showed that two of the trials were not sensitive to additional subjects’ participation while the third trial was quite fragile to one or two additional subjects. Conclusions: The mFI can be a useful metric assessing the fragility of phase I trials and facilitating robust identification of MTD.

## 1. Introduction

The concept of fragility index (FI) dates back to the 1990s [1,2] as an additional robustness appraisal of “statistical significance” for assessing a difference between two proportions. The FI is defined as the minimum number of participants in a randomized clinical trial that is required to change a statistically significant result to non-significant (or vice versa). Walsh et al. used FI to assess the robustness of statistical significance of 399 randomized trials with binary outcomes in 2014 [3], and found that in 53% of trials, the FI was less than the number of patients lost to follow-up, suggesting that the trials were frequently fragile. The FI complements the hypothesis testing (e.g., *p*-value) and helps to identify less robust (or fragile) trial results. Methods for calculating the FI have been further developed for randomized clinical trials with continuous and survival outcomes [4,5], meta-analyses, and network meta-analyses [6,7,8]. The FI has been increasingly applied to many medical fields, including oncology, surgery, obstetrics, and gynecology, during the past decade [9,10,11,12].

In phase I oncology clinical trials, one of the primary goals is to identify the maximum tolerated dose (MTD), which will be used to guide the recommended dose for later phases. However, there may be concerns about the robustness of the chosen MTD, usually determined based on the data from a small number of participants. Researchers and drug developers are often interested in whether the MTD would change if a few additional patients were added to the dose-finding process. If the MTD decision would not be altered in either direction (i.e., downgrading or upgrading) after adding multiple new subjects, we would have more confidence in the dose level chosen as the MTD. Conversely, if the chosen MTD level would be changed right away after including one additional subject, it would raise substantial concerns about whether the selected MTD was reliable. Thus, we propose a modified Fragility Index (mFI), defined as the minimum number of additional participants that is required to potentially change the estimated MTD, to assess robustness and identify fragile phase I trial results.

We begin with an overview on early phase trials and various dose-finding designs in Section 2. Then we cover the definition of mFI and explain how to estimate mFI in Section 3. Three real oncology trials are used to illustrate how to utilize mFI as a convenient and valuable tool for assessing the robustness and validity of the MTD determination in Section 4. Lastly, we examine the relationship between mFI and early stopping, and discuss the limitations of mFI and potential future research topics.

## 2. Materials and Methods

### 2.1. Dose Finding Designs

A phase I trial is often the first time a new drug is applied in human beings. One of the primary goals is to examine the highest possible dose level subject to the dose-limiting toxicity (DLT) constraints and identify the MTD for later phases. Assuming monotonicity, the target DLT probability is often set at a value between 20% and 40%.

The traditional approaches to selecting the MTD include the “3+3” design [13] and various up-and-down designs [14]. The “3 +3” design is the most used for phase I dose escalation. The implementation is easy and does not require a computer program. The sample size required is often smaller than for the model-based designs. However, it is generally inferior in identifying the MTD [15].

Many novel model-based and model-assisted designs have been proposed to improve the efficiency and accuracy of phase I trials to find the MTD. Model-based approaches include the continual reassessment method (CRM) [16], escalation with overdose controls (EWOC) [17], the Bayesian logistic regression model (BLRM), and the time-to-event CRM (TITE-CRM) [18]. Model-assisted designs [19] include the modified toxicity probability interval (mTPI) [15,20], keyboard design [21], and Bayesian optimal interval (BOIN) [22]. Researchers have compared the differences and summarized relative pros and cons for some designs [23]. Appendix A presents an overview of commonly used designs in more detail.

### 2.2. Fragility Index

The FI was developed by Walsh et al. [3] for two-arm randomized controlled trials with binary outcomes that report the numbers of events and non-events in a 2 × 2 frequency table. The aim was to examine whether the statistically significant result of a two-arm trial would be altered with a small change in the number of events. Walsh et al. [3] proposed calculating the FI by changing the event status of a subject in a group with fewer events, re-computing the *p*-value based on the Fisher exact test, and repeating this process until a non-significant *p*-value was reached.

For oncology dose-finding studies, because regulators are often interested in whether the MTD would change if a few additional patients were added to the dose-finding process, we propose a modified Fragility Index (mFI), defined as the minimum number of additional participants required to potentially change the estimated MTD, to assess the robustness and identify fragile trial results. The MTD can be altered in either direction: downgrading to a lower dose level or upgrading to a higher dose level. The aim is to investigate if the MTD decision is sensitive to enrolling a few additional subjects in a phase I dose-finding trial.

Suppose that after a dose-finding trial, we collect the following data: $d=\left(d_{1},d_{2},\ldots d_{J}\right)$ representing the dosage, $n=(n_{1},n_{2},\ldots n_{J})$ the total number of subjects assigned to each dose level, and $y=(y_{1},y_{2},\ldots y_{J})$ the observed DLT for each dose level. If t more subjects are included for further investigation, the possible number of DLTs could be any value in the list {0, 1, 2, …, t}. We could iterate through the list to see whether the originally chosen MTD in the trial would be overturned. As long as one value in the list {0, 1, 2, …, t} changes the MTD decision, the mFI is set to be t and we stop the process. If none of the values in the list {0, 1, 2, …, t} changes the MTD decision, we include one more subject and repeat the same process for the t+1 subjects. If any DLT value in the list {0, 1, 2, …, t+1} changes the MTD decision, we stop the process and conclude that mFI=t+1. To be consistent, it is recommended to use the same dose-finding design as employed in the original trial. However, other dose-finding designs and models can be implemented as supplementary assessments. Figure 1 illustrates the process of obtaining the mFI for a dose-finding trial. The procedure is summarized as follows:
Collect data from a completed dose-finding trial, $d=(d_{1},d_{2},\ldots d_{J})$, $n=(n_{1},n_{2},\ldots n_{J})$, $y=(y_{1},y_{2},\ldots y_{J})$. At the dose level of the MTD ($d_{MTD}$), the number of subjects is $n_{MTD}$ and the number of DLTs is $y_{MTD}$.Start with t=1, i.e., add one additional subject at MTD so that the total number of subjects at the MTD is $n_{MTD}+1$. Let the DLT outcome at the MTD for this new subject be either 0 or 1, hence the total number of DLTs is either $y_{MTD}$ or $y_{MTD}+1$. Use the same statistical method as used in the original study for both numbers of DLTs to see whether the resulting new MTD is different from the original MTD. If it is different, set mFI = 1; otherwise, go to the next step.Let t=t+1. The number of subjects at the MTD is $n_{MTD}+t$ and let the number of DLTs at the MTD take any value between 0 and t: $y_{MTD}$, $y_{MTD}+1,\ldots ,\;y_{MTD}+t$. Use the same statistical method as used in the original study for all DLT outcomes to assess whether the resulting MTD is different. If it is different, set mFI = t; otherwise, go to the next step.Repeat step 3 unless MTD has been changed and mFI has been set to a value, or if it reaches a prespecified large value.Once the mFI value is determined, we can calculate the probability of observing the number of DLTs or a more extreme case that would change the MTD decision based on the estimated toxicity probability at the original MTD level, to assess its likelihood. For example, if t patients are added at the original MTD level and if m or fewer DLTs are observed among those new patients, it will change the MTD; the probability of this happening is:PrX≤m=∑i=0mPrX=i=∑i=0mnMTD+tmpMTDm(1−pMTD)nMTD+t−m
where $X\sim Bin(p_{MTD},\;n_{MTD}+t)$ and $p_{MTD}$ can be estimated using the observed DLT rate at the MTD level.

We have developed a publicly available R package to provide a convenient way to implement the mFI calculation.

## 3. Results: Three Case Studies

### 3.1. Phase I Dose-Escalation Trial of AUY922

A first-in-human phase I trial was conducted in patients with advanced solid tumors to determine the MTD of AUY922 inhibitor [24]. An adaptive Bayesian logistic regression model (BLRM) with overdose control was used to assess the relation between dose and DLT probability. The dose started at 2 mg/m^2^ and upgraded to 4, 8, and 16 mg/m^2^ with no DLTs. At the next higher dose level of 22 mg/m^2^, one DLT was observed among 11 patients. No DLT was seen at 28 mg/m^2^, so the dose was upgraded to 40 mg/m^2^, at which two of the first seven patients experienced DLTs. The BLRM design supported continued dosing at 40 mg/m^2^ on the basis of the assessment that the probability of a true DLT probability above 33% was less than 0.25. Therefore, nine additional patients were then dosed and no DLT was observed among these patients. The dose was further extended to 54 and 70 mg/m^2^, where 2 out of 18 at 54 mg/m^2^ and 3 out of 24 at 70 mg/m^2^ had DLT. The final recommended phase II dose (RP2D) was declared at 70 mg/m^2^. Table 1 displays the dose-finding data of the dose levels, total numbers of subjects treated, numbers of DLTs, and DLT rates for all dose levels.

The mFI was found to be 10, based on both the BLRM and BOIN designs: the MTD did not change until 10 extra subjects were added to the trial at the MTD level of 70 mg/m^2^ and all those 10 subjects experienced DLT. If we tried fewer subjects, the MTD would not be changed no matter how many subjects experienced DLT. This large mFI value suggests that the result for MTD in this trial was quite robust. One can estimate the probability of having a possible number of DLTs, based on the estimated toxicity probability at the MTD dose level, if 10 more subjects were to be recruited onto the trial. Using a binomial distribution, the chance of having 10 DLTs out of 10 additional subjects is very low, $0.125^{10}<10^{-8}$, so it is very unlikely that the MTD result would be altered.

The mFI value was the same with the keyboard design, because both BOIN and keyboard are model-assisted designs and use the same isotonic algorithm to compute the MTD. Other dose-finding designs, such as mTPI, gave the same mFI value of 10, whereas the mFI value was 12 when using the CRM algorithm and the mFI value was 8 when using the EWOC algorithm (Table 2).

### 3.2. Phase I Trial of Pan-AKT Inhibitor MK-2206

This was a dose-escalation study of continuous oral treatment with the pan-AKT inhibitor MK-2206 in patients with advanced tumors [25]. The drug was administered every other day in 28-day cycles to investigate the safety and MTD. In total, 33 patients were dosed at 30, 60, 75, or 90 mg. The dose finding used a two-stage design. In Stage 1, dose escalation proceeded through dose levels of 30 mg (three subjects), 60 mg (three subjects), and 90 mg (seven subjects). There were 4 out of 7 patients who experienced DLTs at 90 mg. In Stage 2, a new dose of 75 mg was introduced for three patients, whereupon all three developed DLTs. An additional three patients were then enrolled at the lower dose level of 60 mg to check the safety parameters of this dose, and no DLTs were found. Stage 2 included 14 more patients in the expansion phase and observed one DLT. The MTD was established at 60 mg with the mTPI design. The dose-finding data for dose levels, total numbers of subjects treated, numbers of DLTs, and DLT rates are displayed in Table 3.

As shown in Table 3, the 100% observed DLT rate at the dose level of 75 mg makes it impossible to upgrade from 60 mg to 75 mg. When additional subjects are enrolled, the only possible outcome of changing the MTD is to downgrade. The mFI is 18 according to the mTPI design algorithm. The high mFI value is caused by the 100% DLT rate at the next dose level. The MTD level does not change until 18 extra subjects are added in the trial at the MTD level of 60 mg/m^2^ and all those subjects experience at least one DLT, the probability of which is extremely low. This suggests that the result for MTD is robust.

As summarized in Table 2, if the keyboard or BOIN design is employed, the mFI is 11, which is still large. Other dose-finding designs give similar mFI values around 10: mFI is 10 with CRM and mFI is 9 with BLRM. However, the mFI is only 5 when using the EWOC algorithm, which may be due to EWOC’s over-conservative safety control rule.

### 3.3. The SPRINT Phase I Trial

SPRINT was an open-label, single arm, multi-center trial of the MEK 1 inhibitor, selumetinib, in children [26]. The phase I portion of the SPRINT trial evaluated three doses of selumetinib, 20 mg/m^2^, 25 mg/m^2^, and 30 mg/m^2^ in pediatric patients, to identify a suitable dose to be used for the next phase based on all available safety, tolerability, pharmacokinetic, and efficacy data. The objective response rate was similar across 20 to 30 mg/m^2^ doses: 66.7% (8/12) at 20 mg/m^2^, 83.3% (5/6) at 25 mg/m^2^, 50.0% (3/6) at 30 mg/ m^2^ respectively. The best rate was observed at 25 mg/m^2^. The tolerability was similar between 20 and 25 mg/m^2^ doses based on the DLT rates: 2 DLTs out of 12 subjects at 20 mg/m^2^, 1 DLT out of 6 subjects at 25 mg/m^2^, and 2 DLTs out of 6 subjects at 30 mg/m^2^, as shown in Table 4. The dose of 25 mg/m^2^ was identified as the MTD and the recommended dose for phase II. We used only the tolerability data listed in Table 4 to assess the robustness of the MTD result.

The mFI was found to be 1 with the BOIN algorithm. When one patient was added with no DLT for this new patient, it led to an upgrade to a higher dose level; and the probability of this happening is 0.833, using the observed DLT rate of 16.7%. On the other hand, we investigated downgrading: first, if only one patient is added and that patient develops DLT, the MTD remains the same; then, if two patients are tested additionally and both two patients develop DLT, it results in downgrading to a lower dose level. The probability of these two patients experiencing DLT is 0.028. The mFI would be the 1, since this is the smallest number of subjects needed to alter the MTD. The small mFI indicates the trial’s MTD conclusion is not robust if we consider only tolerability data. Other dose-finding designs, such as CRM, EWOC, BLRM, and mTPI, give similar mFI values of either 1 or 2 (Table 2).

### 3.4. mFI Results Summary

The mFI results are summarized in Table 2 for all three trials using various dose-finding designs, CRM, EWOC, BLRM, mTPI, keyboard, and BOIN. To be consistent with the original dose-finding process, it is recommended to use the same dose-finding design as employed in the trial for calculating mFI. Other dose-finding models can be implemented as supplementary assessments. There are some variations in the mFI results, as demonstrated by the three trials, but there should be a general pattern or signal in terms of robustness assessment.

The results showed that the first two trials were not sensitive to additional participants, while the third trial is quite fragile to one or two additional subjects being added.

## 4. Discussion and Conclusions

We propose a modified Fragility Index (mFI) to assess the robustness of the MTD determination in early-phase dose-finding trials. The extension of FI in dose-finding trials allows researchers and drug developers to assess whether any additional patients and how many patients should be recruited to achieve a robust MTD decision. Three oncology trials were used to show how to calculate the mFI and assess trial robustness.

In practice, different trials may employ different designs. To be consistent, it is recommended to use the same dose-finding design as employed in the trial for the estimation of mFI. However, other dose-finding designs and models can be implemented as supplementary assessments. Because phase I trials commonly involve a small number of subjects, an mFI value greater than 3 or 5 may intuitively be considered as an indication of robust MTD decision. However, to establish a useful guideline, one would need to systematically evaluate all existing phase I oncology trials and empirically estimate the distribution of mFI. Furthermore, the mFI evaluation may depend on the design used to estimate the MTD. The relative performance of different designs on robust assessment using mFI awaits further research.

Phase I trials can implement rules to stop early if the clinical objectives have been achieved with good confidence. Various stopping rules have been suggested in the literature. O’Quigley et al. [16] proposed a stopping rule based on a confidence interval for CRM and other model-based methods, and Shen and O’Quigley gave a theoretical justification [27]. However, there are some limitations to this approach: (1) the precision level may often require more patients than would be available in an early dose-finding trial and hence, the trial would not halt early in practice; (2) it is questionable whether obtaining a pre-fixed level of precision for the probability of toxicity at the MTD should be a major goal of a trial [28]. On the other hand, O’Quigley and Reiner (1998) [29] estimated the probability that a current recommended dose level will turn out to be the final MTD and the likelihood that all remaining patients will be treated at the current dose level. The trial would be terminated early if the MTD could be predicted with high probability. To implement this stopping rule, one can keep track of the number of times each dose is considered and stop when the dose for an upcoming patient is the same as the dose that was recommended for the previous k (a pre-decided integer) patients in a row. This approach often works out well in practice [28].

Implementing early stopping during a dose-finding trial may first seem quite different from assessing the robustness of a trial using FI, because one is used during a dose-finding trial and the other is considered afterwards. However, they are very much related, because mFI can also be applied in real time during a trial. If a trial has already included a certain number of patients, then what would happen if a few more patients were to be included? Would adding more patients change the current recommended dose? Or would the current recommended dose stay the same, and what probability is associated with that? Therefore, we may want to compare these two approaches via simulations and in practice. As we try to achieve Project Optimus, selecting an optimal biological dose is no longer based just on toxicity, but also other factors, such as efficacy and pharmacokinetic results. Our proposed mFI index can be extended naturally to consider other determining factors beyond safety by incorporating those factors in the decision rule, as shown in the third example.

A related fragility measure is the Robustness Index (RI) proposed by Heston in 2023 [30], which examines how different sample sizes affect statistical significance. When a hypothesis test yields a non-significant result, the original sample size is multiplied by a series of numbers until a significant result is achieved. Conversely, when a test is significant, the original sample size is divided by a series of numbers until the result is no longer significant. The multiplicand or divisor is the RI. However, it is uncertain how the RI can be utilized in a dose-finding trial to determine when the MTD result will change.

The strength of this study is that, to the best of our knowledge, there are scant, if any, existing methods dedicated specifically to evaluating the robustness of results from dose-finding studies. By extending the concept of the FI, this work offers an intuitive way to quantify how the significance of the dose-finding conclusion could change after including additional potential samples. Nevertheless, this study has some limitations. Although our proposed mFI shares the same spirit as the original FI developed by Walsh et al., as both aim at altering the result’s significance, they differ in terms of how they achieve the significance change. Specifically, the original FI was intended for general hypothesis testing in relation to treatment effects with binary outcomes in clinical trials, and it modifies the status of binary outcomes (event or no event), with the size of samples in each group remaining fixed. In contrast, in our proposed mFI, like several other extensions of FI for continuous outcomes and survival outcomes [31], the sample sizes in treatment groups or the study are modified. As such, one may argue that the mFI is not a type of FI measure, and we may consider this as an FI-like measure. In addition, our proposed mFI does not account for the likelihood of DLT outcomes among the assumed additional samples for the mFI calculation. It is possible that the DLT outcomes of some samples may not be clinically practical. Baer et al. proposed generalizing the fragility index to a family of fragility indices called incidence fragility indices, permitting only outcome modifications that are sufficiently likely and providing an exact algorithm to calculate the incidence fragility indices [32]. Such a consideration may also be needed when interpreting the mFI results.

## Figures and Tables

**Figure 1 cancers-16-03504-f001:**
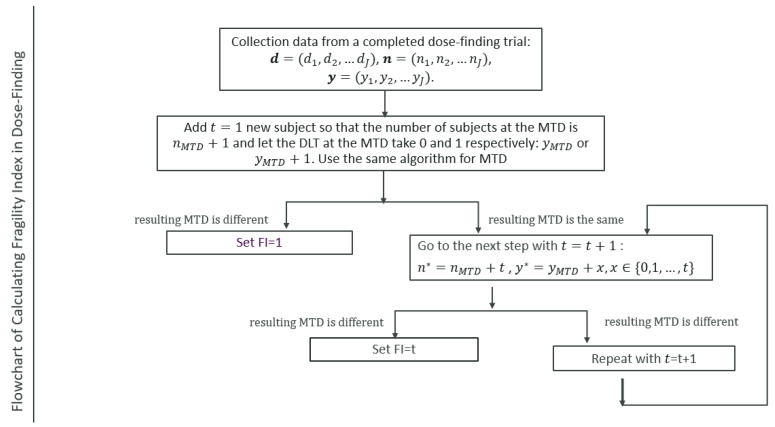
Flowchart of Calculating the modified Fragility Index in a Dose-Finding Trial.

**Table 1 cancers-16-03504-t001:** Summary of the Data in the Phase I AUY922 Dose-Escalation Trial.

	Dose Level Index								
	1	2	3	4	5	6	7	8	9
**Dose level (mg/m^2^)**	2	4	8	16	22	28	40	54	70
**Total # subjects treated**	3	3	4	6	11	8	16	18	24
**# DLT**	0	0	0	0	1	0	2	2	3
**DLT rate (%)**	0	0	0	0	9.1	0	12.5	11.1	12.5

**Table 2 cancers-16-03504-t002:** The mFI Results of Robustness Assessment for All Three Trials.

Trials	Dose-Finding Designs					
	CRM	EWOC	BLRM	mTPI	Keyboard	BOIN
1. AUY922 Dose Escalation	12	8	10	10	10	10
2. Pan-AKT Inhibitor MK-2206	10	5	9	18	11	11
3. SPRINT Trial	2	2	2	1	1	1

**Table 3 cancers-16-03504-t003:** Summary of the Data in the Phase I Pan-AKT Inhibitor MK-2206 Trial.

	Dose Level Index			
	1	2	3	4
**Dose level (mg)**	30	60	75	90
**Total # subjects treated**	3	20	3	7
**# DLT**	0	1	3	4
**DLT rate (%)**	0	5.0	100	57.1

**Table 4 cancers-16-03504-t004:** Summary of the Data in the Phase I SPRINT Trial.

	Dose Level Index		
	1	2	3
**Dose level (mg/m^2^)**	20	25	30
**Total # subjects treated**	12	6	6
**# DLT**	2	1	2
**DLT rate (%)**	16.7	16.7	33

## Data Availability

The summary level data that support the findings of this study are included in the paper.

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
