# Peer review of "Robustness Assessment of Oncology Dose-Finding Trials Using the Modified Fragility Index"

_cancers, 2024, doi:10.3390/cancers16203504_

Round 1
Reviewer 1 Report
Comments and Suggestions for Authors
Thank you for your submission. Your work addresses the important area of fragility, which is crucial for enhancing the reliability of clinical research.
I understand that you state that your proposed mFI has a narrow application (DFTs). However, the concept of assessing robustness through sample size modification has been previously explored in the literature, most notably by Heston (2023) in his work on the Robustness Index (RI) (DOI: 10.7759/cureus.44397).
The similarities between the RI and your proposed mFI are substantial and cannot be overlooked. While the mFI uses addition rather than multiplication (RI), the fundamental concept is nearly identical.
To strengthen your paper and ensure its academic integrity, I strongly recommend the following essential revisions:
1. Include a discussion of the RI compared with the mFI. Clearly delineate how the mFI differs from the RI, both conceptually and in practical application.
2. Provide a robust justification for why in this setting, your method of addition (mFI) may be more appropriate than multiplication using small multiplicands (RI). Or, could either method work?
3. Discuss the specific unique value of your mFI in DFTs compared with the more generalized RI.
These additions should be very straightforward to implement- perhaps a single paragraph in your discussion section would suffice. These revisions will significantly enhance the scholarly value of your paper by situating it within the broader context of statistical fragility measures. Acknowledging prior work in the field is essential for maintaining the academic integrity of your paper.
I look forward to reviewing a revised version of your manuscript that incorporates these essential changes.
Overall, I believe you have done excellent work on an important topic.
Reviewer 2 Report
Comments and Suggestions for Authors
It is always hard to determine how large the number of participants is “enough” for Phase I trials. In particular, determining safe dose amounts requires a lot of participants to generalize the outcomes, but in reality, it is almost impossible. Although there is already the fragility index (FI), this article proposed modifying it to see the significance and determine the minimum number of additional participants required after collecting data from certain numbers of participants. They analyzed the data from three different trials to provide a comprehensive understanding of the proposed modification.
The experiment design is reasonable, and the references are mostly recent ones. I think it is too early to conclude that their method can be used now, but it is good preliminary data at this point. Collecting more data and polishing methods will help establish mFI in the future. Once it is established, mFI will be a powerful tool to enhance the design of many clinical trials with small sample sizes. The potential of mFI to improve future clinical trials is promising. The contents of this article are beneficial for the drug development process, and I recommend publishing this manuscript in Cancers after the following minor revisions.
o In the text, the order of the Tables appeared in Tables 1, 4, 2, and then 3. I understand the authors summarized the data in Table 4, but the readers have to go back and forth when reading. I suggest that you discuss the numbers in each sub-section 3.1, 3.2, and 3.3, and maybe you can add 3.4 as a summary to discuss Table 4.
o According to the guidelines, references are not to be superscript in the text but must be in box brackets [ ].
o Line 18: It may be better to spell out MTD in the Abstract because it is the first time MTD appears in the article.
Here are minor errors.
1) Line 19: The “Method” font is different.
2) Line 27: Needs period before Conclusions
Comments on the Quality of English LanguageMentioned above.
Round 2
Reviewer 1 Report
Comments and Suggestions for Authors
Excellent revision. Thank you.